# Laser Irradiation Synthesis of AuPd Alloy with Decreased Alloying Degree for Efficient Ethanol Oxidation Reaction

**DOI:** 10.3390/ma17081876

**Published:** 2024-04-18

**Authors:** Nan Jiang, Liye Zhu, Peng Liu, Pengju Zhang, Yuqi Gan, Yan Zhao, Yijian Jiang

**Affiliations:** 1School of Physics and Optoelectronic Engineering, Beijing University of Technology, Beijing 100124, China; jiangnan@emails.bjut.edu.cn (N.J.); zly2019@emails.bjut.edu.cn (L.Z.); tianchenhe@emails.bjut.edu.cn (P.L.); z176816@emails.bjut.edu.cn (P.Z.); ganyq@emails.bjut.edu.cn (Y.G.); xishuaibin@emails.bjut.edu.cn (Y.J.); 2Key Laboratory of Trans-Scale Laser Manufacturing Technology, Beijing University of Technology, Ministry of Education, Beijing 100124, China; 3Beijing Engineering Research Centre of Laser Technology, Beijing University of Technology, Beijing 100124, China

**Keywords:** pulsed laser irradiation in liquids, decreased alloying degree, AuPd alloy, ethanol oxidation reaction

## Abstract

The preparation of electrocatalysts with high performance for the ethanol oxidation reaction is vital for the large-scale commercialization of direct ethanol fuel cells. Here, we successfully synthesized a high-performance electrocatalyst of a AuPd alloy with a decreased alloying degree via pulsed laser irradiation in liquids. As indicated by the experimental results, the photochemical effect-induced surficial deposition of Pd atoms, combined with the photothermal effect-induced interdiffusion of Au and Pd atoms, resulted in the formation of AuPd alloys with a decreased alloying degree. Structural characterization reveals that L-AuPd exhibits a lower degree of alloying compared to C-AuPd prepared via the conventional co-reduction method. This distinct structure endows L-AuPd with outstanding catalytic activity and stability in EOR, achieving mass and specific activities as high as 16.01 A mg_Pd_^−1^ and 20.69 mA cm^−2^, 9.1 and 5.2 times than that of the commercial Pd/C respectively. Furthermore, L-AuPd retains 90.1% of its initial mass activity after 300 cycles. This work offers guidance for laser-assisted fabrication of efficient Pd-based catalysts in EOR.

## 1. Introduction

Direct ethanol fuel cells (DEFCs) are regarded as a highly promising green power source, owing to their high energy density and environmental benefits [1,2,3,4]. Nevertheless, slow kinetics at both the cathode and anode impede large-scale application, necessitating advanced catalysts to reduce reaction barriers [5]. Palladium (Pd), known for its ample reserves and moderate cost, is considered the leading anodic catalyst for ethanol oxidation reaction (EOR) [6,7,8]. Nonetheless, the electrocatalytic performance of conventional pure Pd nanomaterials is suboptimal, stemming from Pd’s intrinsic electronic structure [9,10]. To overcome this challenge, alloying Pd with other metals, such as Cu, Fe, Sb, Ni, and Bi, not only bolsters electrocatalytic performance but also curtails Pd usage, resulting in cost savings [11,12,13,14,15,16]. Moreover, the qualities of gold (Au), recognized for augmenting Pd-based catalysts with electronic benefits and poison resistance, have made AuPd alloys the subject of extensive research [17,18,19].

Extensive research demonstrates that catalysts uniformly alloyed via co-reduction display superior catalytic activity [20,21]. Nevertheless, ethanol oxidation reactions predominantly rely on continuous Pd sites for the adsorption and cleavage of reactant bonds [22,23]. Consequently, modulating the alloying degree to alter the prevalence of continuous Pd sites can affect the electrocatalytic efficacy. The laser liquid-phase synthesis of nanomaterials has garnered considerable interest due to its low thermal budget [24], high efficiency, and environmental friendliness [25]. Pulsed laser irradiation in liquids (PLIL), characterized by its pronounced non-equilibrium and quenching effects [26], can rapidly elevate the temperature of nanoparticles to very high temperatures (thousands of kelvins) [27] within an exceedingly brief duration and subsequently cool them back to ambient temperature with a cooling rate ranging from 10^6^ to 10^12^ K/s [28]. This extreme non-equilibrium process facilitates the stabilization of certain unique metastable structures [29]. However, the fabrication of AuPd alloys with a decreased degree of alloying through laser irradiation in liquids requires further investigation.

In this study, we utilized 532 nm nanosecond pulsed laser irradiation in liquids to synthesize a high-performance electrocatalyst of a AuPd alloy with decreased alloying degree (L-AuPd). The structure and composition of the L-AuPd were thoroughly characterized by energy dispersive X-ray spectroscopy (EDS), transmission electron microscopy (TEM), X-ray photoelectron spectra (XPS), X-ray absorption fine structure (XAFS) spectroscopy, X-ray diffraction (XRD), and CO stripping voltammetry. It was shown that L-AuPd presents a lower degree of alloying than the conventionally co-reduced homogeneous AuPd alloy (C-AuPd). The distinctive structure conferred notable catalytic activity and stability on L-AuPd during the EOR. DFT calculations elucidated the factors contributing to the enhanced catalytic activity and stability. Based on the characterization results, we speculate that the formation of the AuPd alloy with a decreased alloying degree is attributed to the combined effects of laser photochemistry and photothermal processes. This work offers guidance for laser-assisted fabrication of efficient Pd-based catalysts in EOR.

## 2. Experimental Methods

### 2.1. Preparation of Au Nanoparticles

Au nanoparticles were produced chemically at room temperature [30]. The precursor solution was prepared by mixing a solution of HAuCl_4_ in ethylene glycol with the same volume of an oleylamine and tetralin blend. Subsequently, 20 μL of the reducing agent of the borane tert-butylamine complex at a concentration of 50 mM was added to the precursor solution to make the two mix well, and the reduction was carried out for 1 h at 20 °C. Finally, 3 mL of alcohol was added to the mixed solution to precipitate the Au nanoparticles, which were subsequently isolated by centrifugation at 9000 rpm for 10 min.

### 2.2. Preparation of L-AuPd

Initially, Au nanoparticles prepared via chemical methods were suspended in a defined volume of oleylamine to create the Au seed solution. Then, 60 mg of ascorbic acid (AA) was mixed with a PdCl_2_ solution with a Pd^2+^ concentration of 0.785 mM, and the mixture was dissolved in ethylene glycol to prepare the Pd precursor solution. The precursor solution and the Au seed solution were mixed well by ultrasonication. Subsequently, the combined solutions were exposed to irradiation from a Nd:YAG laser emitting nanosecond pulses at a 532 nm wavelength. The laser operated with a 10 ns pulse width, 100 mJ pulse energy, a beam spot roughly 0.8 cm^2^ in size, and a 33 Hz repetition rate. Following the laser exposure, 5 mL of alcohol was introduced to the solution to induce precipitation of the L-AuPd nanoparticles, which were subsequently isolated by centrifugation at 9000 rpm for 10 min.

### 2.3. Preparation of C-AuPd

The C-AuPd was synthesized via a conventional chemical co-reduction method. A defined amount of HAuCl_4_ and H_2_PdCl_4_ was added to 10 mL of deionized water, yielding a homogeneous solution. The atomic ratio of Au to Pd in HAuCl_4_ and H_2_PdCl_4_ was set at 1:1. Subsequently, 5 mg of carbon black was introduced to the solution and sonicated for 30 min. Following this, 500 μL of a 50 mM NaBH_4_ solution was added. Two hours later, the mixture underwent washing and centrifugation to yield C-AuPd.

## 3. Results and Discussion

### 3.1. Characterization of the L-AuPd

L-AuPd nanoparticles were synthesized through 532 nm nanosecond pulsed laser irradiation in liquids. Initially, Au nanoparticles, averaging 8 nm in diameter (Appendix A), were produced chemically at room temperature. Then, the as-prepared Au nanoparticles were dispersed into a solution, which contains specific concentrations of Pd^2+^ and ascorbic acid. The solution was irradiated with a 532 nm laser for the preparation of L-AuPd nanoparticles. Please consult the Experimental Methods section for further information on preparation. The HAADF-STEM picture of L-AuPd nanoparticles dispersed on carbon black is shown in Figure 1a. The inset of Figure 1a displays a size distribution histogram for L-AuPd nanoparticles, indicating an average nanoparticle diameter of 3.8 nm. The size of L-AuPd nanoparticles decreases compared to Au nanoparticles. This change is due to photothermal vaporization during laser irradiation [31].

To confirm the composition of L-AuPd nanoparticles, an EDS line scanning measurement was conducted (Figure 1c). The EDS line scanning across the L-AuPd nanoparticles is marked by the dashed line in Figure 1b. The profiles show that the atomic fraction of Au and Pd is intertwined, indicating that both elements are co-present within the nanoparticle. Additionally, EDS elemental mapping analysis, as depicted in Figure 1d, confirms the presence of Pd atoms intermingled with Au, suggesting the alloy’s elemental distribution. In the EDS picture, individual atoms seen surrounding the nanoparticles are attributed to the TEM analysis, which dispersed them from the L-AuPd nanoparticles due to electron beam exposure (Appendix A). The selected area electron diffraction (SAED) pattern depicted in Figure 1e reveals the fcc structure of L-AuPd nanoparticles. Figure 1f displays an AC-HAADF-STEM picture of the L-AuPd nanoparticles. Lattice spacings were measured at 0.231 nm, which lay between that of fcc pure Pd (d _(111)_ = 0.224 nm) and fcc pure Au (d _(111)_ = 0.235 nm), demonstrating the alloy structure of L-AuPd nanoparticles. Figure 1g and the inset are the inverse fast Fourier transform (IFFT) and fast Fourier transform (FFT) results of Figure 1f along the [011] zone axis. These white lines denote the $\left(1\bar{1}1\right)$ and $\left(\bar{1}1\bar{1}\right)$ planes, respectively, demonstrating a distinct twin structure within this region. The orange line indicates the shared crystal face of the twin associated with the $\left(\bar{2}00\right)$ plane. Therefore, these results indicate that the alloy structure of L-AuPd nanoparticles was successfully synthesized via pulsed laser irradiation in liquids.

The preparation parameters of L-AuPd nanoparticles are listed as follows: L-AuPd-3 (100 mJ, 3 min), L-AuPd-6 (100 mJ, 6 min), L-AuPd-9 (100 mJ, 9 min). Figure 2a displays the absorbance spectra in the UV–Vis range for L-AuPd prepared at varying irradiation durations. Near 520 nm, the notable peak in absorption strength is ascribed to the localized surface plasmon resonance (LSPR) phenomenon in Au nanoparticles. As the laser irradiation time increases, the ~520 nm absorption peak broadens and eventually vanishes at the 9th minute. This result indicates that the structure of L-AuPd nanoparticles alters over irradiation time. The XRD patterns in Figure 2b confirm the crystalline phases change with laser irradiation. The XRD peaks of Au nanoparticles are characteristic of the face-centered cubic structure, with the peak at approximately 38.2° associated with the Au (111) crystal plane. For all irradiation times, the diffraction peaks of L-AuPd nanoparticles lie between that of Au (JCPDS No. 04-0784) and Pd (JCPDS No. 46-1043). As irradiation time increases, the most intense (111) peak shifts marginally towards higher angles, indicating the doping of Pd into the Au lattice to form a AuPd alloy. Figure 2c displays the variation in the (111) plane’s lattice spacings as determined by Vegard’s Law compared to those predicted by XRD data. Vegard’s law is a fundamental principle in materials science that describes the relationship between lattice parameters of solid solutions and composition changes in binary systems [32]. The C-AuPd is synthesized using NaBH_4_ in a chemical co-reduction method. The most pronounced (111) diffraction peaks occur at 38.21° (L-AuPd-3), 38.6° (L-AuPd-6), 39.03° (L-AuPd-9), and 39.22° (C-AuPd). Relative to the peak at 38.2° (Au/C), diffraction peaks of the alloys show a positive shift. *d_XRD_* denotes the lattice spacing for (111) crystal surface and is determined as follows:(1)$$2d_{XRD}*\sin\left(\theta \right)=n\lambda .$$

As per Vegard’s law, the lattice spacing of uniform solid solution alloys is intermediate between the constituent metals and is influenced by the composition of each alloy. ICP-OES results for L-AuPd nanoparticles and C-AuPd are presented in Appendix A. The Vegard lattice spacing (*d_Vegard_*) for the (111) crystal surface and the lattice constant (*a*) are calculated as follows:(2)$$d_{Vegard}=\frac{a}{\sqrt{1^{2}+1^{2}+1^{2}}},$$
(3)$$a=\left(1-x_{Pd}\right)a_{Au}+x_{Pd}a_{Pd}.$$

The lattice constant of Pd is expressed as *a_Pd_*, and likewise, the lattice constant of Au is denoted as *a_Au_*. *x_Pd_* is the atomic percentage of palladium in the alloy. Δd denotes the difference between *d_Vegard_* and *d_XRD_*. The details of the calculations are given in Appendix A. For C-AuPd synthesized via the co-reduction method, the Δd value is negative. For L-AuPd nanoparticles, in contrast, the Δd value is positive, increasing initially and then decreasing with prolonged laser irradiation time. The absolute Δd value (|Δd|) of L-AuPd nanoparticles is an order of magnitude larger than that of C-AuPd [33]. Between 3 and 6 min, the Δd increased from 1.45 × 10^−3^ nm to 2.86 × 10^−3^ nm, demonstrating the decrease in alloying degree of the L-AuPd nanoparticles. Simultaneously, the Au:Pd atomic ratio shifted from 1:0.38 to 1:1. We infer that during this time interval, the rate of photodeposition exceeded that of atomic diffusion under the photothermal effect. Between 6 and 9 min, the Au: Pd atomic ratio changed from 1:1 to 1:1.05 with no significant variation, suggesting depletion of Pd^2+^ in the solution by 6 min, resulting in no further reduction and deposition of Pd atoms. Notably, during this time interval, Δd decreased from 2.86 × 10^−3^ nm to 1.31 × 10^−3^ nm, indicating an increase in the degree of alloying. Therefore, we speculate that the degree of alloying increased due to atomic diffusion under the influence of the photothermal effect between 6 and 9 min. The variation in the alloying degree of the AuPd alloy with laser irradiation time is illustrated in the schematic diagram (Figure 2d).

To elucidate the low-alloyed structure of L-AuPd nanoparticles further, we selected the L-AuPd-6 sample with the lowest degree of alloying in L-AuPd nanoparticles as the sample group and the homogeneous C-AuPd prepared via a traditional co-reduction method as the control group for comparative analysis of the XPS results. Appendix A and Appendix A show the XPS results of L-AuPd-3 and L-AuPd-9. Compared with the binding energies (BEs) of the Pd 3d_5/2_ and Pd 3d_3/2_ signals of Pd/C (335.10 and 340.36 eV), those of L-AuPd-6 (335.40 and 340.66 eV) are shifted by about +0.3 eV, while those of C-AuPd (335.04 and 340.30 eV) are shifted by about −0.06 eV (Figure 3a; Appendix A). Meanwhile, the BEs of Au 4f_7/2_ and Au 4f_5/2_ signals of L-AuPd-6 (83.75 and 87.42 eV) and C-AuPd (83.61 and 87.28 eV) both are shifted by about −0.16 and −0.3 eV, respectively, compared with those of Au/C (83.91 and 87.58 eV) (Figure 3b; Appendix A). The different changes in the BEs of Au and Pd in L-AuPd-6 and C-AuPd indicate that L-AuPd-6 and C-AuPd have different surface compositions and structures. For instance, the BEs of Au 4f and Pd 3d of C-AuPd are negatively shifted compared to Pd and Au, consistent with previous studies [34,35,36,37,38]. Due to the alloying effect between Au and Pd, Au gains sp electrons from Pd and loses an almost compensating number of d electrons, with a very small net charge transfer [34,38]. The Au 4f and Pd 3d BEs of L-AuPd-6 are negatively and positively shifted, respectively, which can be attributed to the consecutive Pd sites in L-AuPd-6 leading to the alloying interaction being weakened.

The X-ray absorption near edge structure (XANES) spectra of the Au L_3_ edge of the L-AuPd-6 show a significant increase in the intensity of the second band (11,935 eV) compared to that of the Au foil (Figure 3c). This feature arises from the bimetallic distance effect, which further confirms the gold–palladium alloy phase [39]. The Fourier-transformed k^2^-weighted EXAFS spectra shown in Figure 3d reveal three characteristic peaks of Au foil at 2.15, 2.52, and 2.94 Å, while for L-AuPd-6 and C-AuPd, the peaks at 2.15 and 2.85 Å correspond to Au-Au and Au-Pd bonds, respectively [40]. The incorporation of Pd changes the triple peaks of Au to a doublet due to the scattering from Au-Pd bonds [41]. Quantitative local structures were obtained by fitting the FT-EXAFS spectra. The quality of the fit is shown in Figure 3e and Appendix A, and the fitted structural parameters are summarized in Appendix A. The fitting results indicate that the first shell layer’s average coordination numbers (CNs) for C-AuPd are marginally greater than those for L-AuPd-6, and the second shell layer’s CNs for C-AuPd are slightly less than those for L-AuPd-6. It is evident that the coordination environment around the L-AuPd-6’s gold atoms has changed, with some continuous Pd sites emerging in contrast to C-AuPd [39]. This indicates a change in the degree of alloying in L-AuPd-6. To elucidate the causes of these variations, we conducted a wavelet transform (WT) analysis on the EXAFS maps, affording a more distinct representation of R-space and k-space. Figure 3f illustrates that the profile intensity of Au foil peaks at (9.37 Å^−1^, 2.75 Å), whereas for AuPd alloys, the maximum intensities occur at (6.5 Å^−1^, 2.15 Å) and (4.85 Å^−1^, 2.85 Å), attributable to Au-Au and Au-Pd bonds, respectively. The differences in peak positions in the WT graphs between L-AuPd, C-AuPd, and Au foil demonstrate the alloying of Au with Pd to form AuPd alloys [42]. Compared to C-AuPd, L-AuPd-6 shows greater dispersion intensity for Au-Au bonds, consistent with the fitting results, indicating a change in the alloying degree of L-AuPd-6 [43]. Therefore, based on the characterization results, it can be concluded that L-AuPd NPs have a lower alloying degree than C-AuPd NPs.

### 3.2. Electrocatalytic Performance for EOR

The EOR performance of L-AuPd nanoparticles prepared with varying time parameters is shown in Appendix A. L-AuPd-3 exhibits the optimal catalytic performance, with a mass activity of 20.20 A mg_Pd_^−1^ and a specific activity of 35.83 mA cm^−2^—11.54 and 9.05 times higher than commercial Pd/C. However, its stability is poor, just like a “firework”. L-AuPd-9 possesses excellent stability, maintaining 92.5% of its mass activity after 300 cycles; nevertheless, its catalytic performance is subpar. Therefore, considering both catalytic activity and stability, L-AuPd-6—sharing the same atomic ratio as C-AuPd—was selected for comparison with C-AuPd and Pd/C. In a 1.0 M KOH and 1.0 M ethanol solution, cyclic voltammograms (CVs) indicate that L-AuPd-6 demonstrates significantly higher current density than both C-AuPd and Pd/C (Figure 4a). As depicted in Figure 4b, L-AuPd-6 achieves a mass activity of 16.01 A mg_Pd_^−1^ and a specific activity of 20.69 mA cm^−2^, substantially surpassing C-AuPd’s 5.66 A mg_Pd_^−1^ and 12.24 mA cm^−2^, and is 9.1 and 5.2 times greater than commercial Pd/C. Furthermore, L-AuPd-6 has exhibited the highest recorded mass activity among Pd-based catalysts reported in the literature (Figure 4d and Appendix A). The electrochemical surface area (ECSA) of L-AuPd-6 is 50.20 m^2^ g_Pd_^−1^, exceeding that of C-AuPd (29.54 m^2^ g_Pd_^−1^) and Pd/C (28.66 m^2^ g_Pd_^−1^) (Appendix A). The Tafel slopes presented in Figure 4c correspond to the EOR kinetics of the catalysts [44]. L-AuPd-6 demonstrates the fastest reaction kinetics with a slope of 158 mV dec^−1^, surpassing C-AuPd (180 mV dec^−1^) and Pd/C (244 mV dec^−1^). As depicted in Figure 4e, for L-AuPd-6, the CO oxidation onset potential is as low as that for C-AuPd, and even lower when compared to Pd/C, demonstrating that AuPd alloys facilitate CO removal at lower potentials compared to Pd/C. The peak of CO oxidation for C-AuPd experienced a negative shift of 15 mV compared to Pd/C, suggesting a reduced CO binding strength on C-AuPd [45,46]. For L-AuPd-6, the peak of CO oxidation showed a positive shift of 15 mV when compared to commercial Pd/C, which indicates a stronger CO binding. In the existing literature [47], it has been observed that the CO oxidation peak undergoes a positive shift in Au@Pd core-shell NPs due to the enhanced CO binding affinity caused by the presence of continuous Pd sites. This finding further validates the presence of analogous continuous Pd sites in the low-alloyed L-AuPd-6. However, the exposure of Au atoms in the low-alloyed L-AuPd-6 allows for the retention of continuous Pd sites while concurrently harnessing the synergistic Au-Pd effect. Consequently, this enhances both the efficiency of CO oxidation removal and the dehydrogenation of ethanol, thereby distinguishing L-AuPd-6 from Au@Pd core-shell NPs.

Chronoamperometry (CA) measurements were performed to evaluate the durability of the catalysts. The curves were recorded at 0.77 V vs. RHE. L-AuPd-6 maintains a significantly higher mass activity throughout the test compared to Pd/C and C-AuPd, both of which show rapid decay (Figure 4f). This result indicates the superior long-term mass activity of L-AuPd-6 toward EOR. Accelerated durability test (ADT) results reveal slower activity decays in L-AuPd-6 during cycling, with its residual catalytic activity remaining at 90.1%, outperforming C-AuPd at 83.7% and Pd/C at 55.8%, reflective of L-AuPd-6’s enhanced stability. Following the ADT, L-AuPd-6 remained well dispersed on carbon black, with slight agglomeration in some areas (Appendix A), which indicates that L-AuPd-6 has excellent structural stability, enabling it to maintain high catalytic activity after 300 cycles.

### 3.3. DFT Calculations

Density functional theory (DFT) calculations were performed to further investigate the effect of the continuous Pd sites on EOR. The atomic model of the continuous Pd sites in L-AuPd catalysts is shown in Figure 5a, along with the atomic models of C-AuPd and pure Pd. To investigate the electronic structure of the catalysts, the projected density of states (PDOS) for Pd atoms on the surfaces of L-AuPd, C-AuPd, and pure Pd are calculated, as shown in Appendix A. The result indicates that the d-band center of L-AuPd is at −1.37 eV, situated between that of C-AuPd (−1.30 eV) and pure Pd (−1.62 eV). Notably, the PDOS intensity near the Fermi level is significantly reduced for both L-AuPd and C-AuPd due to the interaction between Au and Pd. L-AuPd exhibits a higher PDOS near the Fermi level than C-AuPd. The depletion of d-band electrons near the Fermi level can lead to weaker adsorption of intermediates [48,49]. To further investigate the adsorption strength of intermediates on the catalyst surface, the free energy of L-AuPd, C-AuPd, and pure Pd were calculated (Figure 5b). The result shows that C_2_H_5_OH adsorption is 0.035 eV on L-AuPd and 0.045 eV on C-AuPd, which both are endothermic, but on pure Pd (−0.075 eV), it is exothermic. Notably, the adsorption energy of C_2_H_5_OH on C-AuPd is higher compared to that on L-AuPd, demonstrating the C_2_H_5_OH adsorption process of L-AuPd was improved. In Appendix A, State2, the energy barrier for the first dehydrogenation step, is smaller on L-AuPd (0.078 eV) compared to C-AuPd (0.083 eV). In the next dehydrogenation step with the formation of CH_3_CHO* (Appendix A, State 3), the reaction energy barrier on C-AuPd is −0.01 eV, indicating that the reaction is exothermic. Nevertheless, for L-AuPd and pure Pd, the endothermic reaction serves as the rate-determining step, with the energy barrier for L-AuPd (0.007 eV) being lower than that for pure Pd (0.16 eV). In the third exothermic dehydrogenation step (Appendix A, State 4), L-AuPd displays an energy barrier of 0.32 eV, nearly equivalent to that of C-AuPd (0.306 eV). The energy barrier for the formation of CH_3_COOH in L-AuPd is 0.054 eV, which is almost comparable to 0.042 eV for C-AuPd, and both are significantly lower than 0.071 eV for pure Pd. The results suggest that the low-alloyed structure with consecutive Pd sites on L-AuPd is more favorable for the dehydrogenation process of ethanol, thereby improving its oxidation performance.

CO_ads_ is a common poisonous intermediate during EOR, and OH_ads_ plays a key role in CO_ads_ removal [50]. Figure 5c illustrates that the adsorption energy of CO on L-AuPd is −1.28 eV, similar to that on C-AuPd (−1.27 eV) and significantly lower than that on pure Pd (−1.38 eV). Meanwhile, the adsorption strength of OH on L-AuPd is more pronounced, at −2.73 eV, compared to C-AuPd (−2.41 eV) or pure Pd (−2.50 eV). A comparison of CO and OH adsorption energies between L-AuPd and C-AuPd reveals that L-AuPd, with its continuous Pd sites, can mitigate the adsorption of the toxic intermediate CO and concurrently facilitate CO removal through augmented OH adsorption [51]. Consequently, L-AuPd demonstrates superior CO anti-poisoning efficacy and outstanding stability.

## 4. Conclusions

In summary, we successfully synthesized a high-performance electrocatalyst of a AuPd alloy with a decreased alloying degree via pulsed laser irradiation in liquids. Comprehensive structural characterization revealed that L-AuPd has a lower alloying degree than conventionally co-reduced homogeneous C-AuPd. This distinct structure endows L-AuPd with exceptional catalytic activity and stability in EOR, achieving mass and specific activities as high as 16.01 A mg_Pd_^−1^ and 20.69 mA cm^−2^, 9.1 and 5.2 times that of the commercial Pd/C, respectively. Furthermore, L-AuPd retains 90.1% of its initial mass activity after 300 cycles. DFT calculations further indicate that compared with C-AuPd, the low-alloyed L-AuPd effectively lowers reaction energy barriers for ethanol adsorption and CH_3_CHOH* formation and enhances CO anti-poisoning capability by augmenting OH* adsorption. Based on the characterization results, we speculate that the formation of the AuPd alloy with a decreased alloying degree is attributed to the combined effects of laser photochemistry and photothermal processes. This work offers guidance for laser-assisted fabrication of efficient Pd-based catalysts in EOR.

## Figures and Tables

**Figure 1 materials-17-01876-f001:**
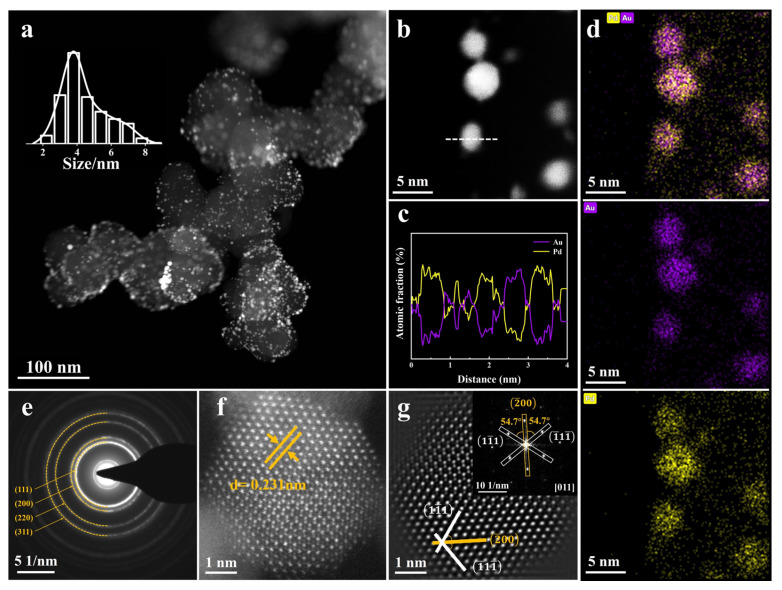
(**a**) HAADF-STEM picture of L-AuPd nanoparticles/C; inset displays size distribution histogram. (**b**) HAADF-STEM picture of L-AuPd nanoparticles. (**c**) EDS line scanning across an L-AuPd nanoparticle, as marked by the dashed line in (**b**). (**d**) Elemental profiles from HAADF-STEM-EDS, (**e**) SAED analysis, (**f**) AC-HADDF-STEM picture, (**g**) IFFT representation of (**f**); inset depicts the FFT of L-AuPd nanoparticles.

**Figure 2 materials-17-01876-f002:**
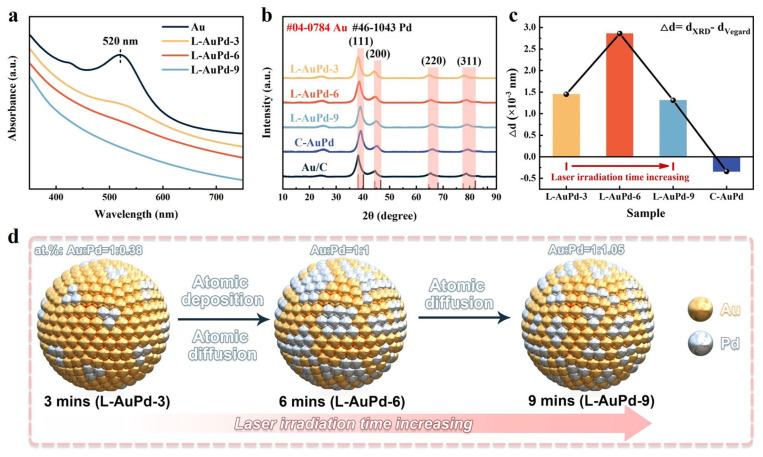
(**a**) UV–Vis absorbance spectra for L-AuPd nanoparticles. (**b**) XRD patterns for L-AuPd nanoparticles, C-AuPd, and Au/C. (**c**) Disparity in (111) plane lattice spacings from XRD data versus Vegard’s law. (**d**) Schematic depicting variation in L-AuPd nanoparticles’ degree of alloying with laser irradiation time.

**Figure 3 materials-17-01876-f003:**
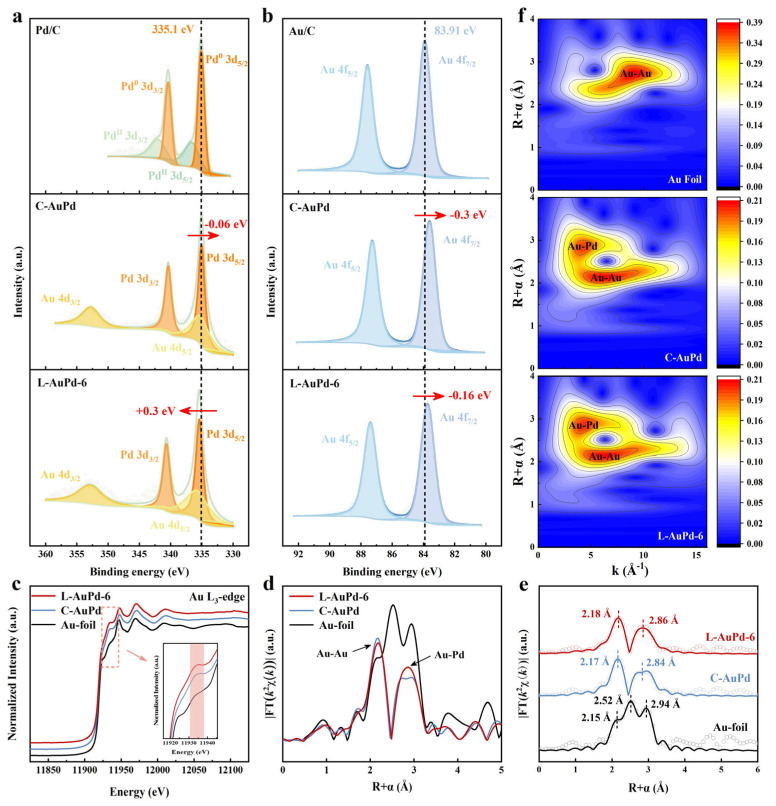
(**a**) XPS spectra of Pd 3d signals of Pd/C, C-AuPd, and L-AuPd-6. (**b**) XPS spectra of Au 4f signals of Au/C, C-AuPd, and L-AuPd-6. (**c**) Au L_3_ edge XANES spectra, (**d**) the Fourier transform k^2^-weighted EXAFS spectra, (**e**) k^2^-weighted Au L_3_ edge FT-EXAFS spectra, as well as the fitting curves and (**f**) wavelet transform for EXAFS spectra of Au foil, C-AuPd, and L-AuPd-6.

**Figure 4 materials-17-01876-f004:**
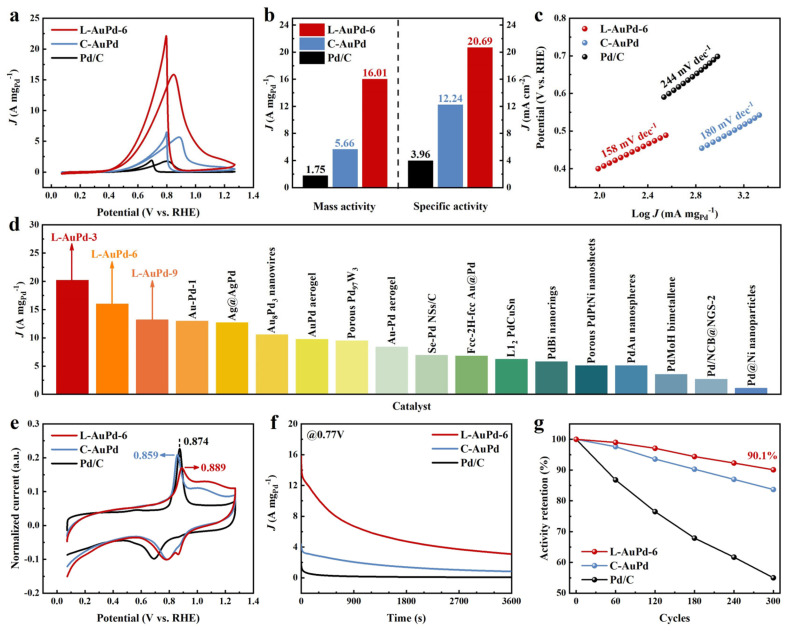
(**a**) CV profiles for Pd/C, C-AuPd, and L-AuPd-6 were obtained in N_2_-saturated 1.0 M KOH and 1.0 M ethanol at room temperature, with a scan rate of 50 mV s^−1^. (**b**) Mass activities and specific activities for the catalysts. (**c**) Tafel plots for the catalysts. (**d**) Comparative analysis of mass activity between L-AuPd catalysts and other reported Pd-based EOR catalysts. (**e**) CO stripping profiles for the catalysts in 1.0 M KOH. (**f**) Chronoamperometric responses for the catalysts were recorded at 0.77 V vs. RHE in 1.0 M KOH and 1.0 M ethanol. (**g**) Potential cycling stability of the catalysts in 1.0 M KOH and 1.0 M ethanol for 300 cycles with a scan rate of 50 mV s^−1^.

**Figure 5 materials-17-01876-f005:**
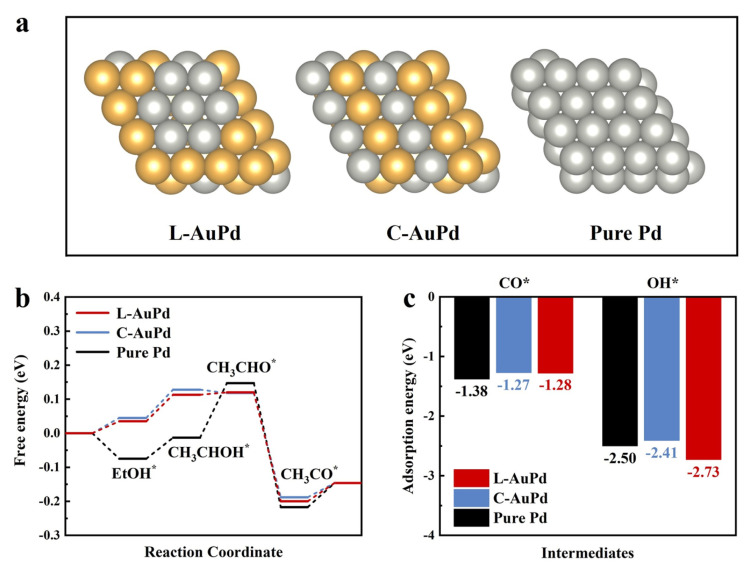
(**a**) Atomic model. (**b**) EOR free-energy profiles and (**c**) adsorption energies of OH* and CO* species for pure Pd, C-AuPd, and L-AuPd.

## Data Availability

Data are contained within the article and Appendix A.

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
