# Peer review of "Laser Irradiation Synthesis of AuPd Alloy with Decreased Alloying Degree for Efficient Ethanol Oxidation Reaction"

_materials, 2024, doi:10.3390/ma17081876_

Round 1
Reviewer 1 Report
Comments and Suggestions for Authors
The paper is very interesting to read and the authors have performed a detailed characterisation. While the Laser irradiation methods and the Pd-Au based catalysts were well reported to dominate the commercial Pd/C catalysts, I would suggest to add the literature based on laser methods in the introduction.
The authors have reported the comparison of Pd based catalysts by various methods and represented the data in Table S5. I would recommend adding the synthesis method against each catalyst reported as a separate column .
If possible , please add the data based on laser based methods in Table S5 as that would allow a direct comparison of the laser methods rather than other synthesis methods.
Reviewer 2 Report
Comments and Suggestions for Authors
Authors reported Lasar irradiation synthesis of AuPd alloy with decreased alloying degree for efficient EOR. Authors performed several electrochemical and physicochemical characterizations and explained well. This manuscript can be accepted for publication after following revisions:
1. Author should mention how they decreased alloying degree of AuPd in abstract section.
2. The rational design of bimetallic alloy can be elaborated in introduction with the reference of following article: doi.org/10.1016/j.ijhydene.2019.06.170
3. The explanation for XAS is insufficient. Authors should index the peak positions in figures also.
4. How about the electrochemically active surface area of the prepared catalysts?
5. Please update the revised manuscript with PDOS profile also.
Comments on the Quality of English LanguageMinor editing of English language required
